# Freedom to Grow: Improving Sow Welfare also Benefits Piglets

**DOI:** 10.3390/ani11041181

**Published:** 2021-04-20

**Authors:** Orla Kinane, Fidelma Butler, Keelin O’Driscoll

**Affiliations:** 1Teagasc Pig Development Department, Animal & Grassland Research and Innovation Centre, Moore-Park, Fermoy P61 P302, Ireland; Keelin.ODriscoll@teagasc.ie; 2School of Biological, Earth and Environmental Sciences, University College Cork, Cork T23 TK30, Ireland; F.Butler@ucc.ie

**Keywords:** animal welfare, pig production, farrowing crate, free lactation, piglet performance

## Abstract

**Simple Summary:**

This study identified positive effects on piglet welfare and growth rates when housed in free lactation pens compared with conventional farrowing crates. Free lactation pens allow piglets unrestricted interaction with the sow and increased space allowance; this improved behavioural development of piglets thus resulted in fewer instances of damaging behaviour toward littermates in this study. Piglets also benefit from better access to the udder for feeding in free lactation pens, with those piglets reared in such pens having better growth rates than those in conventional farrowing crates.

**Abstract:**

Piglet mortality, especially due to crushing, is a concern in pig production. While current systems use farrowing crates to reduce mortality, they present major animal welfare problems for the sow. This study investigated the effects of free lactation farrowing accommodation on the welfare of piglets born and reared in such accommodation when compared with conventional farrowing crates. Piglets were born to sows in one of two farrowing accommodation treatments and followed from birth to slaughter. Treatments were conventional farrowing crates (control, *n* = 24 litters) and free lactation pens, which were larger and allowed the sow freedom of movement both pre and post farrowing, (free, *n* = 22 litters) (675 total piglets). Individual weights were recorded from birth to slaughter. Hoof score was recorded at weaning. Behaviour was recorded during lactation and during the weaner stage. Pre-weaning percentage mortality was equal in treatments (free = 15.95 ± 2.31, control = 14.42 ± 2.15, *p* = 0.61). Final weight was influenced by treatment (*p* < 0.05) with pigs from free lactation pens significantly heavier at 114.73 kg compared to 110.82 kg for control pigs. Free pigs took fewer days than control pigs to reach the target weight of 105 kg (147.6 vs. 149.23 days). Throughout lactation, fewer instances of damaging behaviour (ear and tail biting) were observed in free pigs (*p* = 0.07). Improved growth rates and a decrease in damaging behaviours provide evidence to suggest that pigs from free lactation pens experience improved welfare when compared with control pigs.

## 1. Introduction

The farrowing crate was designed to reduce piglet mortality by restricting the movements of the sow and allowing the piglets a safe space to retreat [1,2]. However, farrowing crates can have a negative effect on welfare by physically restricting sows, thus compromising their behaviour and comfort [2,3,4]. Moreover, even though they were developed to protect piglets, there are conflicting results regarding piglet survival rates compared with alternative lactation housing that allow the sow greater freedom of movement; Kilbride et al. [5] and Weber et al. [6] reported no difference in mortality between crates and loose pens, whereas Blackshaw et al. [7], and Marchant et al. [8] reported greater mortality when sows are loose-housed in pens. Piglet mortality is influenced by a range of factors other than housing systems, including litter size at birth, and the age or parity of the sow [9]. When Weber et al. [6] compared piglet mortality in loose pens and farrowing crates they found no difference in total piglet mortality; 1.40 piglets/litter lost in loose systems and 1.42 piglets/litter in farrowing crates. However, they also considered the cause of mortality, and reported more crushing in loose housing (0.62 piglets/litter) than farrowing crates (0.52 piglets/litter). Thus, overall mortality was equal due to significantly higher mortality from causes other than crushing in crates (0.89 piglets/litter) when compared with loose housing (0.78 piglets/litter). This demonstrates the complexity of the issue, and how factors which improve some aspects of animal welfare, may have a negative impact on others. Mortality due to crushing usually occurs within the first 4 days after birth, Marchant et al. [8] reported over half of liveborn mortality to occur at this time.

Piglets reared in alternative farrowing systems, such as free farrowing crates, have been reported to weigh more at weaning than those reared in conventional crates [10]. Alternative systems often have a greater space allowance than conventional pens, and the sow has freedom to self-select a lying location when she has freedom of movement. Thus, in pens where the sow has greater freedom of movement, piglets may have improved access to the udder. Indeed, milk let down lasts longer and fewer piglets miss milk let down in alternative farrowing pens, compared to traditional closed crate pens [10].

Post weaning weight, and weight at finish, are extremely important key performance indicators. As loose pens have been shown to improve pre-weaning growth rates, they are also associated with heavier weaning weights. Weaning weight is an important predictor of performance, with heavier weaning weights associated with improved growth rates, and reduced days to slaughter [11,12]. Thus, if the improvement in growth pre-weaning is maintained post-weaning, the use of free lactation crates may offer production advantages to the producer, as well as benefits to sow and piglet welfare; however, to date, the long term implications of free lactation housing systems are under-researched.

Besides performance, housing systems which allow the sow and piglets greater freedom of movement and space can also affect other measures of welfare. Increased space allowance and social contact, which are associated with loose housing, are generally recognised as important to improving animal welfare [13]. One aspect of piglet behaviour which could be affected by increased space and social contact is the performance of play behaviour. This has been shown to be very important in pigs, as they are highly motivated to perform such behaviour and it can have effects on social and cognitive development [14]. Thus, environments which promote, rather than limit, play behaviour may result in improved welfare [15].

The effects of an improved physical and social early life environment may also last beyond the time that pigs are managed there. For instance, piglets reared by loose-housed sows showed lower levels of damaging behaviour as well as more play behaviour post weaning than piglets reared by confined sows in a study by Webster and Dawkins [16], which suggests that the lactation environment has a significant effect on the behaviour of pigs later in life. It is possible that providing the opportunity for increased interactions with the sow may improve the development of social behaviours which could make piglets better able to adapt to the stress of weaning [17].

Hoof health is another indicator of welfare and is important for physical comfort. Mouttotou and Green [18] found the prevalence of sole bruising to be 50% in a study of 264 pre-weaning piglets on 13 farms in England. In most cases, claw injuries are superficial with no evidence of an effect on performance. However, if damage is severe, it becomes a welfare issue and can also lead to the development of infection and lameness. It is, therefore, important that alternative farrowing systems do not negatively affect piglet hoof health.

Many studies of chronic stress across a range of species report changes in cortisol concentration [19,20,21] and thus cortisol levels are commonly used as a marker of stress in pigs and other animals [22,23,24]. A problem in assessing animal welfare is that the collection of samples may in itself be stressful for the animal; therefore, non-invasive methods are recommended both for animal welfare purposes, and to improve the quality of the data [25]. Faecal samples offer the advantage that they can be easily collected without causing stress to the animal [26]. They may also provide a more long-term insight into an animal’s cortisol levels than a ‘snapshot’ measure, such as blood or salivary cortisol levels, as the amounts detected are likely to be representative of a longer period. Faecal cortisol has not yet often been measured in studies of piglet welfare and may be a useful tool in monitoring welfare on commercial farms.

The aim of this study was to determine whether piglet welfare and performance is improved in free lactation pens compared to pens where sows are managed in conventional farrowing crates. Specifically, the study aimed to identify whether piglet behaviour, growth, hoof health, and cortisol levels differed when sows were managed in free lactation pens, compared with conventional farrowing crates.

## 2. Materials and Methods

This study was carried out at Teagasc Pig Development Department in Moorepark, Fermoy, Co. Cork, Ireland. Experimental work was authorized by the Teagasc animal ethics committee (Approval no.: TAEC192-2018).

Four farrowing batches (26–30 sows/batch) were used in the experiment. From each batch, 12 sows which were in good general health and showed no signs of clinical lameness were selected for the study (*n* = 46 sows in total), at day 108 of gestation (day—1 of the experiment; D-1). This was the day prior to movement from gestation housing to the farrowing pens (D0).

Within each batch, sows were assigned to one of six blocks on the basis of locomotion score, as described in [27], (1.5 ± 0.51 (1–2)), parity (2.57 ± 2.01 (1–6)), teat number (15.15 ± 1.15 (14–18)) weight (275.69 ± 39.85 (188–358)), and back-fat thickness (17.02 ± 3.63 (10–26)). One sow from each block was then randomly assigned to one of the treatments: control or free (i.e., six sows per treatment per batch). Treatment pens were located in one of three farrowing rooms. One room contained six free pens. Two other rooms contained seven control pens. Within each batch, only one of the control rooms were used, and only the 6 control sows within that batch were located in the room (i.e., the 7th farrowing pen was left empty).

The control treatment consisted of conventional farrowing crates which were installed in farrowing pens measuring 184 × 250 cm (4.6 m^2^) (Figure 1). The free treatment consisted of a similar crate, located in a larger pen (212 × 261 cm, 5.5 m^2^). In the control treatment, the crate confined the sow and allowed for very little freedom of movement; allowing the sow to stand and lie, but not to turn or move around the pen. The crate in the free pens allowed for the sow to be confined as before, yet the crates could also be opened to allow the sow an increased level of freedom of movement (Figure 1). When the crate was opened, the sow could freely turn around through 360°.

Once in the farrowing rooms, sows in the control treatment were confined in the crate from entry until weaning, a period of five weeks. In the free treatment, the farrowing crates were initially left open so that sows were loose and able to turn around in the pens. From the afternoon of D5, the crates were closed (16:00) to confine the sows overnight and to allow for habituation to the crate, then opened again each morning (08:00). When sows in the free crates were observed to be producing milk (an indication that they were close to farrowing) the crate remained closed in the morning. The free sows remained confined from first sign of milk until day 4 post farrowing. After this period of confinement, the crate was reopened, and the sow was allowed freedom of movement until weaning. Farrowing was not induced in either treatment.

Farrowing rooms were artificially lit from 07:00 to 16:30 h. Sows were fed using a computerised feed delivery system (DryExact Pro, Big Dutchman, Vechta, Germany). Sows were fed twice daily from D0 (entry to the farrowing room) to day 14 of lactation and three times daily thereafter until weaning. The sow lactation feeding curve started at 2.9 kg/d at day 0 of lactation and gradually increased to 6.3, 7.8, 8.7 and 8.2 kg/d on average, at days 7, 14, 21 and 26 of lactation, respectively. Feed troughs were checked once per day in the morning to assess sow feed intake and individual feeding curves were adjusted accordingly, by increasing or decreasing the feed allowance by 5%. Water was provided on an ad libitum basis to sows from a single-bite drinker in the feed trough and to suckling piglets from a bowl in the farrowing pen. Farm staff were present on the farm from 07:00 to 16:30 h each day to assist with farrowing, provide general care to the animals, and administer medication if necessary. One sow was removed from the trial due to a shoulder lesion. A total of twelve piglets, from across both treatments, were removed from the trial due to health and welfare reasons, such as hunger (as evidenced by failure to thrive) or injury; these piglets were moved to a nurse sow and were not reintroduced to the trial.

All piglets were tagged, weighed and their sex determined within 24 h of birth. Cross fostering was carried out where necessary, to ensure litter number did not exceed functional teat number, within 48 h of birth, a final number of 675 piglets (355 control in 24 litters; 320 free in 22 litters) remained on the experiment. The average number of suckling piglets in each litter in the free treatment was 14.64 ± 1.47, and in the control treatment was 14.79 ± 1.61. Piglets were never cross fostered more than once. Piglet’s teeth were clipped and, for all but two pens of piglets (one from each treatment), tails were docked during the first 48 h. All piglets received an iron injection on D5 post-partum and males remained fully intact. Creep feed was introduced at approximately 14 days of age. Enrichment in the form of hessian sacks and small natural fibre plant pots were introduced at approximately 12 days of age. Daily records of any mortalities and their cause were kept.

At weaning (26.5 ± 1 days of age) piglets were moved as entire litters without re-mixing to weaner pens which measured 2.4 m × 2.6 m; and had fully slatted plastic floors. The maximum number of pigs in a pen was 12 (i.e., 0.52 m^2^/pig); in litters where more than 12 pigs were weaned, piglets which were lame or underweight were removed from the experiment at this point. Enrichment in the form of a rubber floor toy (Easyfix Luna 117^®^, Easyfix, Ballinasloe, Ireland) and a wooden spruce post was provided in every pen.

Pigs were moved to finisher pens (4 m × 2.4 m; 9.6 m^2^) approximately 7 weeks post weaning and again remained in intact groups. Enrichment in the form of one hanging rubber chew device (Easyfix Astro 200^®^ Easyfix, Ballinasloe, Ireland) and a spruce wooden post were provided in each pen. Pigs spent 9 to 12 weeks in the finisher house before being sent to slaughter.

The temperature was maintained at 28 °C immediately post-weaning in the weaner house by automatic heating and mechanical ventilation. Thereafter it was lowered by 2 °C every 2 weeks. In the finisher house, mechanical ventilation maintained a temperature of 20 °C. Rooms were equipped with windows, which enabled the pigs to experience natural light. Artificial lighting (150 lux in weaner house and 130 lux in the finisher house) was provided for 10–12 h/day to ensure sufficient lighting to retain a normal circadian rhythm.

In both the weaner and finisher stage, pigs were fed ad libitum via a single spaced wet-dry feeder with dry pelleted feed, with a nipple drinker providing water. Feed supply was managed via a computerised feed system (DryExact Pro, Big Dutchman, Vechta, Germany).

### 2.1. Experimental Measures

#### 2.1.1. Mortality

All cases and causes of mortality were monitored throughout the trial. In the farrowing rooms, dead piglets were examined for crushing injuries (traumatic injuries, e.g., bruises or visible impressions of the slatted floor on the piglet’s body). Removal from the trial for other reasons such as hunger or injury were recorded. In the weaner and finisher stage, cause of death and/or reason for removal from the trial were determined by experienced farm staff.

#### 2.1.2. Pre-Weaning Measures

##### Weight

Piglets were individually weighed within 24 h of birth, on day 7, day 14, and day 21 of age, and at weaning (26.5 ± 1 days of age). These data were used to determine the litter weight at each weighing, and piglet pre-weaning average daily gain (ADG).

##### Behaviour

Piglet behaviour was recorded 4 times per day on day 8, day 15, and day 22 of age by direct focal sampling for 3 min, twice in the morning (09.00–12.00) and twice in the afternoon (12.00–15.00). The observer stood outside the back wall of the pen and recorded all occurrences of locomotory play, social, object directed and damaging behaviours, according to the ethogram described in Table 1.

##### Faecal Cortisol

Piglet faecal samples were collected on one day per week, at approximately 5 (4.58 ± 0.90), 12 (11.58 ± 0.90), 19 (18.58 ± 0.90) and 26 (25.58 ± 0.90) days of age. A fresh composite faecal sample (approximately 5 g), uncontaminated with urine, was collected from multiple locations in the dung pile in the corner of the pen, and placed in a sealed plastic vial, then stored at −20 °C until analysis. Prior to analysis, 5 mL of 80% methanol was added to 0.5 g faeces and centrifuged at 2500 rpm for 15 min as described in Palme et al. [28]. At analysis samples were defrosted, centrifuged and analysed in duplicate using a commercially available salivary cortisol assay kit (Expanded range high sensitivity salivary cortisol enzyme immunoassay kit, Salimetrics Europe Ltd., Suffolk, UK), according to the manufacturer’s procedure. Inter and intra-assay CV were 32.2% and 8.8%, respectively.

##### Hoof Scores

The condition of piglets’ hooves was assessed at weaning. All four hooves were examined and individually scored using a scoring system adapted from Lewis et al. [29], Table 2.

#### 2.1.3. Post-Weaning Measures

Only pens with 11 or 12 pigs at weaning (13 free pens and 12 control pens) were included in post-weaning analysis, in order to control for the effect of space allowance on performance and behaviour measurements.

##### Live Weight and Performance

Pigs were individually weighed unfasted at move from the weaner house to the finisher house and at weekly intervals once they approached slaughter weight (beginning on week 9 of the finisher stage). At each of the weigh days, in the finisher stage, pigs that weighed over 105 kg were sent for slaughter the next morning. On week 12 after entry to the finisher stage, all remaining pigs were sent for slaughter, regardless of reaching the target weight.

Feed quantity delivered to each pen in the weaner and finisher stage was downloaded daily from the computerised feed system. These data were used to calculate the average daily feed intake (ADFI) at pen level until the recording day that the first pigs went to slaughter. Combined with pen weights at weaning, the move to the finisher house and at the first slaughter date, average daily gain (ADG) and feed conversion efficiency (FCE; ADFI/ADG) for both weaner and finisher stages were calculated.

##### Behaviour

Pigs were subjected to behaviour tests on days 8, 15 and 22 after the move to the weaner pens. A one-week delay in commencing these tests was observed to ensure pigs were habituated to their new environment. A total of four tests were carried out in the same order on each testing day in order to standardise the testing procedure, adapted from Welfare Quality 2009 [30] and previously described by Schmitt et al. [30]

First, a startle test was used to measure reaction to a sudden event, and the capacity of the pigs to recover. The observer opened an umbrella over the pen and immediately started a timer. The startle reaction of the pigs was scored as follows; 1 = at least 60% of pigs in the pen were startled, or 0 = less than 60% startled. ‘Startled’ was defined as the pigs stopping whatever activity they had been engaged in and being immobile for at least one second. In startled groups, the latency for the group to return to normal behaviour after the startle was recorded.

Immediately after the startle test, the pigs’ reaction to a novel object was observed by placing a 20 L water bottle in the middle of the pen and recording the latency for the first pig to make contact with the object. If no pigs made contact with the object within 3 min, the test was terminated.

After the novel object test two human animal relationship tests (HART) were conducted. The first part measured the group reaction toward the presence of a human and the second part measured the response of each individual pig to human contact. For the first test (HART1) the observer entered the pen and scored the panic response of the group as follows; 0 = less than 60% show panic response, fleeing or facing away from the human, and 1 = over 60% show panic response [30]. The second test (HART2) was carried out immediately after HART1 [29,30]. Any pigs which showed fear to human approach and contact, were scored 1, and pigs accepting human contact were scored 0. The experimenter was familiar to the pigs having handled them regularly from birth.

The final test assessed pigs’ willingness to exit the pen and explore a new area, the corridor. Immediately following the HART tests, the door of the pen was opened approximately 30 cm by the experimenter, who remained still and silent. Pigs were free to exit the pen for up to 3 min, after which the test was terminated. The latency to first exit and number of pigs outside the pen at 1 min, 2 min and 3 min were recorded.

#### 2.1.4. Statistical Analysis

Analyses were performed in SAS 9.4 (SAS Institute, Cary NC, USA) available online: https://support.sas.com/software/94/ (accessed on 19 April 2021). Results were deemed statistically significant when α level was below 0.05, and a significant tendency was considered when α level was between 0.05 and less than or equal to 0.1. Either the Tukey–Kramer or Bonferroni adjustments were used for multiple comparisons where least squares means (LS means) were determined and *p*-values were adjusted. Degrees of freedom were estimated using Kenwood–Rogers adjustment. For data that were analysed using general linear models, data are presented as least square means and standard errors. The univariate procedure was used initially for evaluating data distribution and identifying any outliers. Two sows and their litters were excluded from all analysis due to removal from trial for a shoulder injury and savaging of piglets.

##### Pre-Weaning Measurements

The percentage mortality and the percentage of piglets weaned (inverse relationship) were analysed using general linear models (Proc Mixed). For this analysis, the parity of the sow was categorised as 0 (*n* = 10), parity 1 and 2 (*n* = 16), parity 3 and 4 (*n* = 9) or parity 5 and 6 (*n* = 13). The model included the fixed effects of treatment, parity category, the interaction, and replicate. The number of piglets in the pen after cross fostering was used as a covariate.

As well as this, the cause and timing of death were examined for pens that had at least one piglet die. Causes were defined as crushing, hunger and euthanized. The day of death was log transformed so that residuals had a normal distribution. The effect of treatment on both the cause and day of death was analysed using the same linear mixed model. Treatment, cause of death, the interaction, parity, and replicate were included as fixed effects. Fisher’s exact test was also carried out for each of the causes of death, to determine whether there was a difference in rates of death due to each cause before or after day 4 (i.e., when the crate was open in the free pens) across treatments.

For analysis of piglet weights, the sow was used as the experimental unit, and two models were used. The first model investigated weights on each recording day. The model included the fixed effects of treatment (control vs free), whether the mother was primi or multiparous (Gilt vs Sow), the day of weighing (D7, D14 and D21), all interactions between these factors, as well as the experimental replicate. Birth weight and the number of piglets in the litter were included as covariates. Day of weighing and piglet were included as repeated effects, with a direct product compound symmetry structure included to account for covariance between piglets and over time. The second model was used to analyse both weaning weight and average daily gain to weaning. The same fixed and repeated effects were used as before, but with the exclusion of day. The number of days between birth and weaning was included as an additional covariate. A compound symmetry covariance structure was specified.

Pre-weaning behaviours were expressed as the percentage of piglets in the litter performing the behaviour at each observation time. The percentage of piglets resting, and at the udder was analysed, as well as the main behaviour categories in the ethogram (locomotory play, social, exploratory behaviour and negative behaviours). As well as this, the sub-categories of piglet directed and sow directed social behaviour, interaction with enrichment, tail and ear biting behaviours, and fights were considered. The data for each day were averaged over the four recording periods. For interaction with enrichment, only data from the second and third recording days were used, as the enrichment materials had not yet been placed in the pens at the first recording day. Data were analysed using a general linear model (PROC MIXED). Fixed effects included treatment, recording day, the interaction, and replicate. Recording day was included as a repeated measure with an Autoregressive covariance structure. Damaging behaviour had 0.01 added to each value, to account for 0 values, and was then log transformed.

Piglet faecal cortisol levels were also analysed using a general linear mixed model (Proc Mixed). Fixed effects were the treatment, sample day, the interaction, whether the mother was a gilt or multiparous, and the replicate. Elisa plate was included as a random effect, and sampling day a repeated measure.

Piglet hoof scores were analysed using a generalised linear model (Proc Genmod). Fixed effects were treatment and whether the mother was a gilt or multiparous, as well as replicate.

##### Post-Weaning Measurements

For post-weaning weights, only pens containing 11 or 12 pigs were analysed to ensure no confounding of treatment by the number of pigs in pen; in total, 12 control pens and 13 free pens were included in the analysis. The pen was considered the experimental unit. Two models were used; the ADG, ADFI, weight at the end of each stage (for finishers up to the day that the first pigs were sent to slaughter), and the FCE in the weaner and finisher stages were analysed using the first model. Fixed effects included treatment, stage and the interaction, as well as replicate, and the number of pigs in the pen. Stage was included as a repeated effect, with a compound symmetry covariance structure. The second model was used to compare the days to slaughter (from birth) and the weight at which pigs went to the factory. In this model, fixed effects were the treatment and replicate, and the number of pigs in the pen.

The pen reaction to the startle test (i.e., whether the pen showed a startle response or not) was analysed using a generalised linear model (Proc Glimmix). Only test results from the second and third tests were analysed, as on the first test all pens had a startle response. Fixed effects included the treatment and the replicate. The latency to return to normal was analysed using a general linear model (Proc Mixed). Fixed effects included the treatment, test day, the interaction and the replicate. Test day was included as a repeated effect.

Latency to make contact with the novel object was analysed using a similar model to that described above. By the third test day, all pens had a latency of under 6 s, and 9 in total had a latency of 0 s, and as such these data were removed from statistical analysis.

The HART1 was analysed using a generalised linear mixed model, with each test day analysed separately as for the startle test. The HART2 test was analysed using a general linear model, again with the same fixed and repeated effects as for the latency to touch the novel object.

The latency for pigs to exit the pen was log transformed so that residuals approached normality. Data were analysed using a general linear model (PROC MIXED). Fixed effects included treatment, recording day, the interaction, and replicate. Recording day was included as a repeated effect.

## 3. Results

### 3.1. Pre-Weaning Measurements

#### 3.1.1. Mortality

There was no effect of treatment on the percentage mortality or percentage of piglets weaned, or on the day that they died (Table 3). In total, 125 piglets died prior to weaning.

However, there was an effect of parity category on both percentage mortality and percentage piglets weaned (*p* < 0.001 for both; Figure 2). Sows of parity 5 and 6 had significantly higher mortality than those of parity 0 (i.e., gilts; *p* < 0.001) and the cluster of sows of parity 1 and 2 (*p* < 0.01). Sows of parity 3 and 4 also tended to have a higher mortality level than gilts (*p* = 0.06).

Of the piglets that died, the exact cause of death was known for 109. Mortality data and details of the causes of death before and after opening of the crate (morning of day 4 after farrowing) are outlined in Table 3 and Table 4. More piglets were killed due to crushing in the free treatment after the crate was opened than in the control. Although, numerically, more piglets in control died due to hunger after day 4, the difference was not significant.

#### 3.1.2. Piglet Growth

There tended to be an interaction between treatment and day on pre-weaning live-weight (*p* = 0.08), with piglets from the free treatment heavier, although not significantly, on day 14 and day 21 after birth (Figure 3).

There was no effect of treatment on average daily gain during lactation (free = 0.429 ± 0.007 kg/day, control = 0.233 ± 0.007 kg/day; *p* = 0.13) or on total litter weight weaned (free = 86.29 ± 0.84 kg, control = 86.47 ± 0.86 kg; *p* = 0.86). Nevertheless, individual weaning weight tended to be higher in free pigs, 7.83 ± 0.19 kg, than control pigs, 7.40 ± 0.18 kg (*p* = 0.12).

#### 3.1.3. Behaviour

In the free treatment, piglets were observed more often at the udder (*p* < 0.05; Figure 4) than those in the control group. Social, exploratory and locomotory play behaviours were not affected by treatment.

With regard to damaging behaviour, there was no effect of treatment on fighting, and although there was an interaction between treatment and observation day (*p* < 0.05), there was no difference on any individual day, or pattern over time (numerically more in control on day 15, and in free on day 22). However, there tended to be fewer instances of damaging behaviour (ear and tail biting combined) observed in free pigs (0.037 instances/pig/3 min) than in control (0.054 instances/pig/3 min; *p* = 0.07).

#### 3.1.4. Faecal Cortisol

Overall, there was no effect of treatment on faecal cortisol concentrations (free = 0.553 ± 0.159 µg/dL, control = 0.386 ± 0.148 µg/dL; *p* = 0.14). Nevertheless, there was an interaction between sampling day and treatment (*p* < 0.01; Figure 5). On the first sampling day, which corresponded to approximately 1 day after the crate was opened in the free treatment, piglets in free tended to have higher faecal cortisol levels than those from control (*p* = 0.07).

#### 3.1.5. Hoof Scores

There was no effect of treatment on piglet hoof scores at weaning (*p* = 0.57), median (interquartile range) for piglets from both treatments 4 (2–6).

### 3.2. Post-Weaning Measurements

#### Pig Performance

Post-weaning mortality was greater in pigs that had been reared in the control treatment (3.66%) than in the free (2.81%), although this was not statistically analysed due to the low numbers involved.

Pigs reared in the free treatment prior to weaning had a higher ADG overall (0.829 ± 0.014 g/day) than those in in the control (0.782 ± 0.013; *p* = 0.01). This difference was significant in the weaner stage (*p* < 0.05) although not in the finisher stage (Figure 6A). There was no effect of treatment on ADFI either overall or in either stage (Figure 6B). Overall, pigs from the free treatment tended to have better FCE (1.87 ± 0.03 g/g) than those from the control (1.94 ± 0.02 g/g; *p* = 0.07), tending to be significant in the weaner stage (Figure 6C; *p* = 0.09), but not in the finisher stage.

Regarding entire pen weights, pigs from the free treatment performed better. They tended to be heavier than their control counterparts at move from weaner to finisher stage (37.5 ± 1.5 vs. 33.7 ± 1.4; *p* = 0.1) and were significantly heavier by the date when the first pigs were sent to slaughter (109.1 ± 1.6 vs. 103.5 ± 1.4; *p* < 0.05). Moreover, pigs from the free treatment took fewer days to reach the target slaughter weight of 105 kg (147.6 ± 0.5) than those from control (149.2 ± 0.5; *p* < 0.05) and weighed more at slaughter (114.7 ±1.2 kg) than control pigs (110.8 ± 1.0 kg; *p* = 0.01).

### 3.3. Behaviour Post-Weaning

There was no effect of lactation housing on the startle, human approach, or open door test outcomes. However, overall, free pigs took longer to approach the novel object than control (*p* < 0.05). There was also an interaction between treatment and test day (*p* < 0.05); on the first test day (i.e., one week post weaning) pigs from the free treatment took longer to approach the object than those from control (*p* < 0.05), whereas there was no difference on the second test day (Figure 7).

## 4. Discussion

This study shows that free lactation pens which temporarily confine the sow can deliver the same level of piglet survival as farrowing crates, even in litters where there are more than 14 piglets. Moreover, pigs reared in free lactation pens performed better across the entire production cycle, taking fewer days to reach target weight, and finishing at a heavier weight than those reared in farrowing crates. Therefore, this management system may mitigate the sow welfare problem caused by farrowing crates, while also improving their grower pig performance.

Piglet mortality rates were similar in both treatments. The findings of this study are supported by previous studies where overall piglet mortality rates were found to be similar in conventional crates and alternative farrowing accommodation [2,5,31,32]. This was the case even though litter sizes in the current experiment were in general larger than in previous studies, with the number of functional teats matched to the number of piglets. This strategy was utilized so that the results could be relevant to real life situations, as the ongoing increases in litter size mean that space on the udder must be optimized.

One of the primary negatives associated with loose housing systems is that there is generally a higher risk of crushing (and as a result increased overall mortality). However, the strategy of opening crates on day 4 post-farrowing, and thus imposing a ‘free lactation’ management system rather than ‘free farrowing’ appeared to mitigate many of the risks of the system. This is likely because, as also acknowledged in the literature, the first three days post farrowing is the period of highest risk for crushing of piglets [8]. The time of day of crate opening can also influence the risk of crushing; King et al. [33] found mortality to be lowest in crates opened in the afternoon rather than in the morning. In the current study, crates were opened in the morning due to farm management schedules, and as such, it is possible mortality could have been reduced further if opening takes place in the afternoon.

There is no doubt that individual attention to each lactating sow can have an impact on mortality rates. King et al. [33] reported mortality to be highest when all crates were opened once the average litter age was 7 days, and suggested opening crates individually, as we did in the current study. Moreover, large litters with small piglets are particularly vulnerable; Goumon et al. [34] found that litter size influenced weight gain in piglets, with larger litters having decreased weight gain. Thus, while opening crates on day 4 for smaller litters with larger, stronger piglets may work, a longer confinement may be beneficial where litters are larger, and piglets within the batch could have a lighter average birth weight.

It is important not to overlook factors other than crushing which influence mortality; this is a complex issue with individual sows having differing maternal characteristics that can also influence piglet survival. For instance, Hales et al. [35] found that mortality increased with increasing parity, which is in line with our findings, as well as with increasing litter size. King et al. [36] found that the sows’ previous experience in a farrowing system affected piglet mortality. Thus, older sows in this study which were housed in the free treatment may have achieved lower rates of mortality had they previously experienced rearing piglets in a free lactation system. Although not statistically significant, numerically fewer piglets from the free treatment died from hunger or had to be euthanized. This likely compensated for the increase in crushing after the crates were opened, the result of which was similar overall mortality levels pre-weaning.

The finding that numerically fewer piglets from the free treatment died from hunger than those from the crate is in line with the results regarding pre-weaning performance. The interaction between treatment and the days that piglets were weighed prior to weaning showed that piglets from the free treatment became increasingly heavier than those from the control. This translated into a tendency for a heavier weaning weight, and greatly improved post-weaning performance. There are several factors that could have contributed to this. Piglets may have had improved access to the udder due to increased space around the sow, and indeed piglets were observed at the udder more often in the free treatment than the control. Although not measured in this study, milk let-down has been found to last 1.8s longer in free farrowing pens [10]. In that study, piglets had fewer teat fights and fewer piglets missed milk let-down in free farrowing pens compared to control, both factors which are advantageous for growth. Other studies have also found piglets reared in pens where the sow is not confined throughout lactation to be heavier at weaning than those reared in pens with conventional farrowing crates [37]. These results suggest that this type of management, providing the sow with an increased level of freedom pre-farrowing and during lactation not only has benefits for sow welfare, but also for piglet welfare. Free lactation pens were larger than control pens in this study (5.5 m^2^ vs. 4.6 m^2^). This increase in space allowance during lactation may also have had additional positive impacts. Further research into the economic effects of a reduced number of farrowing places due to the increased pen size of such free lactation crates is of importance; as such, the data generated in this study will be incorporated into the Teagasc Pig Production Model [38] so the full impact of such a strategy can be evaluated. As well as no negative effect on mortality and production, there must also be no economic losses associated with this management system if producers are to consider it.

As stated above, pigs reared in the free treatment had not only a tendency for improved growth pre-weaning, but improved performance post-weaning, particularly in the weaner stage. The greater average daily gain, combined with a lack of difference in feed intake, resulted in an improved feed conversion efficiency. Moving to weaner stage accommodation can be a stressful transition due to changes in environment, feed and separation from the sow, ultimately leading to a reduction in feed intake. Therefore, any improvement in ADG at this stage may indicate that pigs are less stressed and is likely to be advantageous with regard to production.

Overall pen weights from the free treatment tended to be greater than those from the control at move from the weaner stage to the finisher stage and were significantly greater 9 weeks after entry to the finisher stage (when the first pigs went to slaughter). This difference in weight at finish may offer an improvement in profitability and could indicate higher welfare. Days to slaughter was significantly reduced in pigs reared in the free lactation treatment, again benefiting production with pigs reaching a heavier finish weight over a shorter period without any increase in feed intake.

As well as being observed more often at the udder, piglets from the free treatment tended to perform fewer damaging behaviours. Although this difference was not significant, a reduction in damaging behaviour suggests that these piglets experienced reduced levels of stress; ear and tail biting in pigs are abnormal behaviours associated with stress [39]. Martin et al. [15] found that piglets born in alternative farrowing pens displayed play behaviour sooner and played more during the pre-weaning stage than those born in crates. The same study found that free farrowing crate piglets displayed less damaging behaviour post-weaning. Ursinus et al. [40] suggests that tail biters likely stem from litters where tail biting is common across the litter; thus, our finding that biting behaviour tended to be reduced in the free treatment predicts this type of lactation accommodation may have benefits for pigs later in life.

Although there was no effect of treatment on faecal cortisol concentrations, there was a tendency for higher cortisol levels for piglets in the free treatment. This period coincided with the initial opening of the crate. This could indicate increased stimulation or stress when the crate is first opened as piglets interact more freely with the sow and the sow becomes more active in the pen. Indeed opening of the crate has been associated with short-time increased restlessness or stress in sows, as well as increased chasing of piglets [34], which could increase the stress experienced by the piglets. As samples were collected at pen level, this may also be a result of faeces from piglets which could have been mildly injured by the sows increased movement included in the sample. However, the lack of a difference in hoof lesions suggests that there was at the least no difference in injuries to the legs due to sow movement.

In the post-weaning behaviour tests, pigs reared in free lactation pens took longer to approach a novel object. However, this was only in week one, and by the second week post weaning there was no difference. One limitation of the test could, however, be that we used the same novel object both weeks, which may have led to it losing novelty. This suggests a higher level of caution in free pigs, which is a normal behaviour. Schmitt et al. [41] found that pigs reared in a poor pre-weaning environment (an artificial rearing enclosure from day 7 post farrowing until weaning) expressed less fearful behaviour in similar post-weaning tests. The authors hypothesised that in that case, these animals became more habituated to human contact than sow-reared piglets, as they had a smaller enclosure and fewer locations to hide. It is possible that there were similar underlying causes in this study. In free pens, it was slightly more difficult to catch piglets for handling, and as such, there may have been more of what could be perceived as ‘negative’ interactions between piglets and humans; as piglets could escape into the larger sow lying area, removal for procedures such as weighing was a more prolonged activity than in the pens with standard crates. This could have caused piglets to become more wary of unexpected events. It is important to consider that these results may not be representative of conditions on a commercial pig farm; the study was conducted in a research unit where the pigs were very familiar with human contact and were subjected to more frequent procedures such as weighing.

## 5. Conclusions

Rearing pigs in free lactation pens as described in this study had positive implications for post weaning performance, with final weight and days to slaughter improved compared with conventional farrowing crates. The performance data suggest that these pigs were better equipped to deal with the stress of weaning, which had a long-lasting positive effect on growth, increased space allowance pre-weaning may have also contributed. Pre-weaning damaging behaviour was reduced through the use of free lactation pens which is an important finding as the industry moves towards rearing pigs with intact tails. Most importantly for both welfare and production, mortality rates were not affected by using free lactation pens compared with traditional farrowing crates. Changes to management such as confinement for a longer period depending on litter size may further minimise losses. Thus, free lactation crates hold potential to ease a transition to a complete removal of crating of sows, as they can minimise losses for producers, and enhance growth, while meeting more of the needs of the animals than traditional farrowing crates.

## Figures and Tables

**Figure 1 animals-11-01181-f001:**
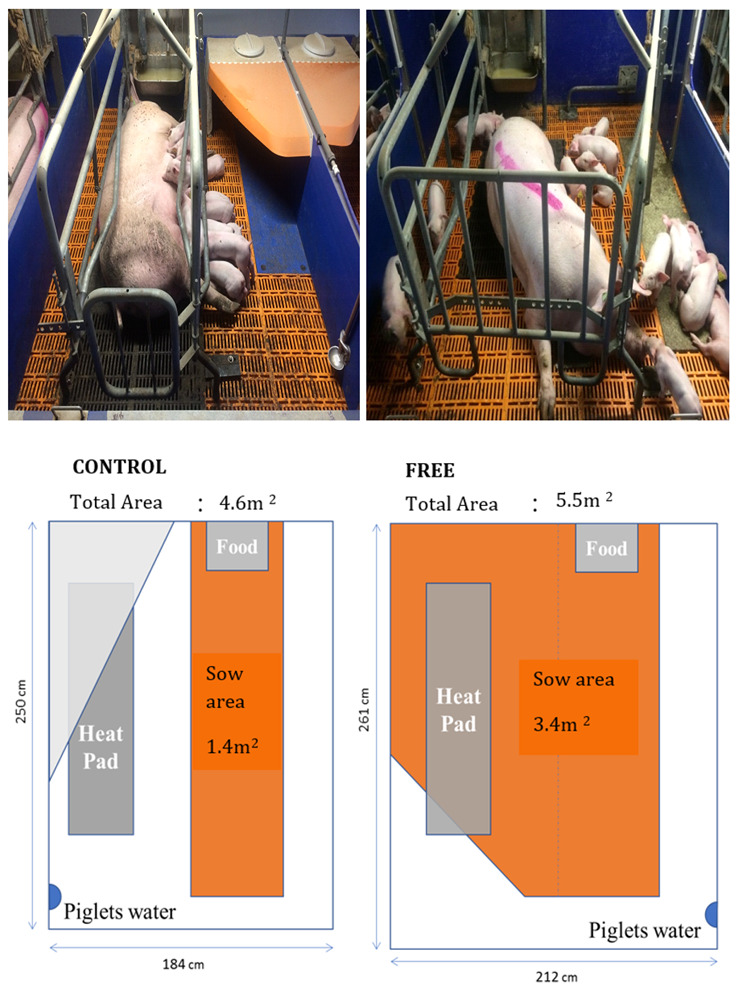
Photographs and schematics (not to scale) of the pens used in the control (standard farrowing pen) and free lactation pens used in the experiment.

**Figure 2 animals-11-01181-f002:**
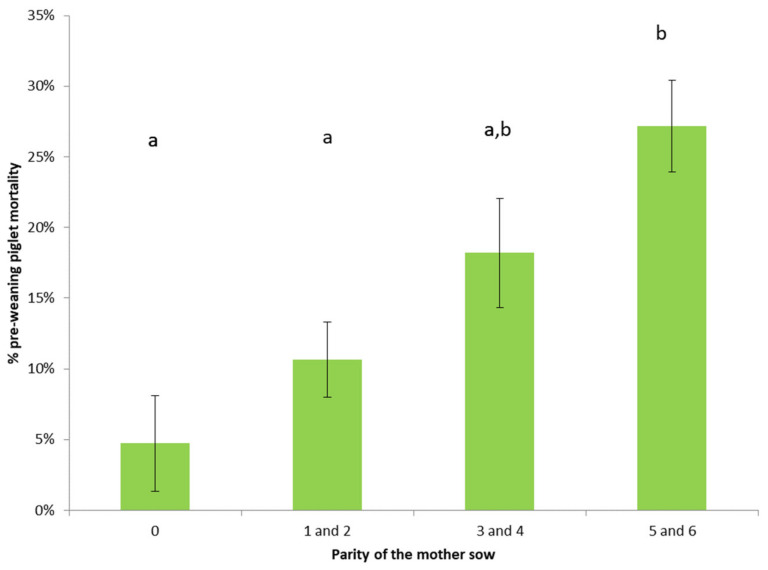
Percentage mortality prior to weaning in sows of various parities. (a, b) indicate a significant difference at *p* < 0.01.

**Figure 3 animals-11-01181-f003:**
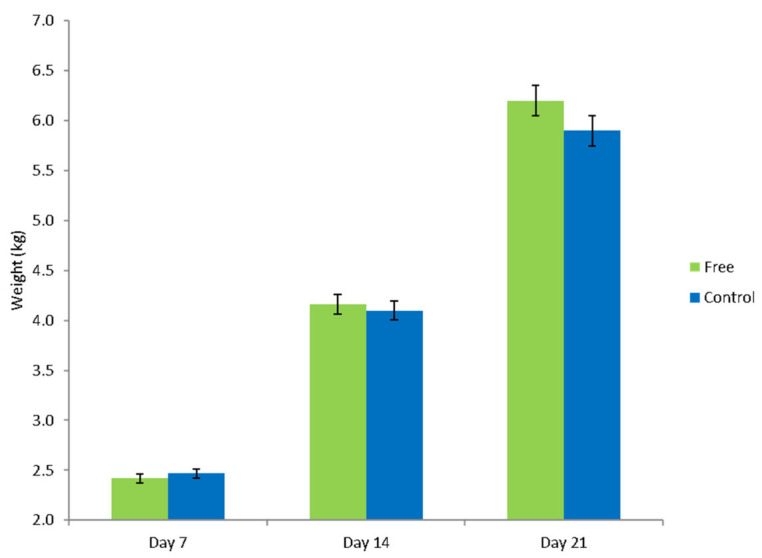
Piglet weights on days 7, 14 and 21 after birth, for piglets born and reared in free lactation pens (free) and conventional farrowing crate pens (control).

**Figure 4 animals-11-01181-f004:**
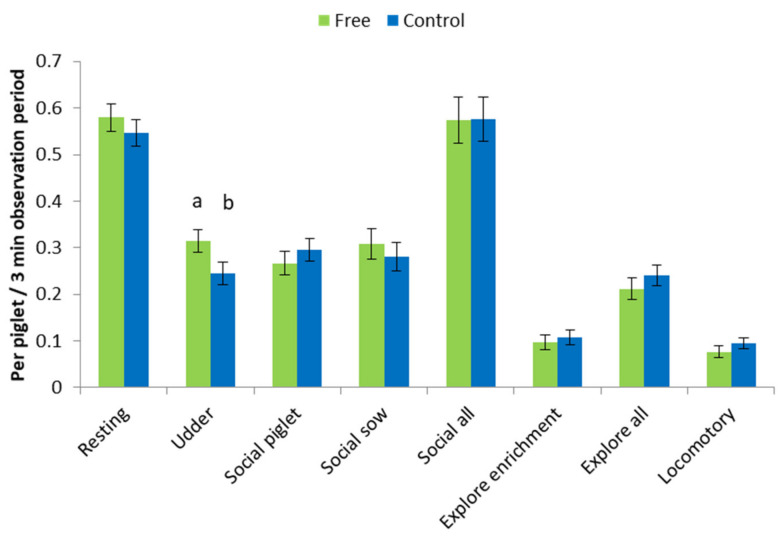
Piglet behaviours during lactation for piglets in free pens where the sow was free to turn around from day 4 of lactation to weaning (free) and conventional farrowing crates (control). ^a, b^ indicates a significant difference at *p* < 0.05.

**Figure 5 animals-11-01181-f005:**
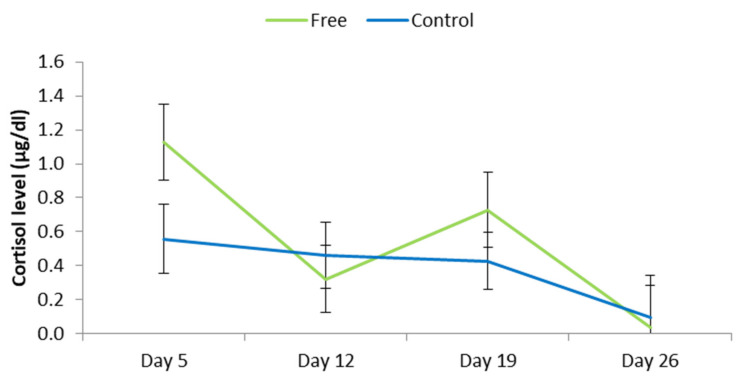
Faecal cortisol levels of piglets in free and control treatments during lactation. Free piglets were managed in pens where the sow was free to turn around from day 4 of lactation to weaning (free) and control were in conventional farrowing crates.

**Figure 6 animals-11-01181-f006:**
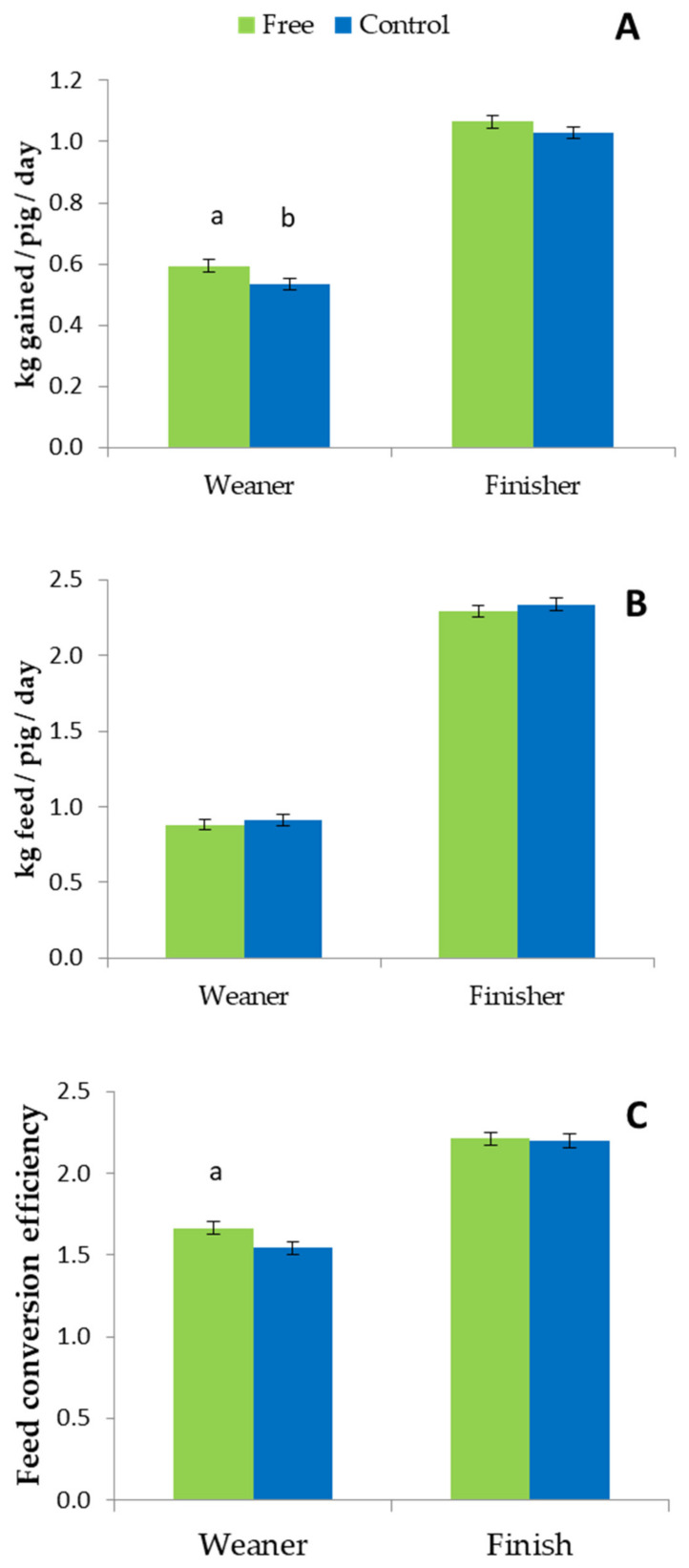
(**A**) Average daily gain (ADG), (**B**) Average daily feed intake (ADFI) and (**C**) Feed conversion efficiency (FCE) for pigs from free and control treatments post-weaning. Free and control treatments during lactation. Free pigs were managed in pens where the sow was free to turn around from day 4 of lactation to weaning (free) and control were in conventional farrowing crates. ^a, b^ indicates a significant difference at *p* < 0.05.

**Figure 7 animals-11-01181-f007:**
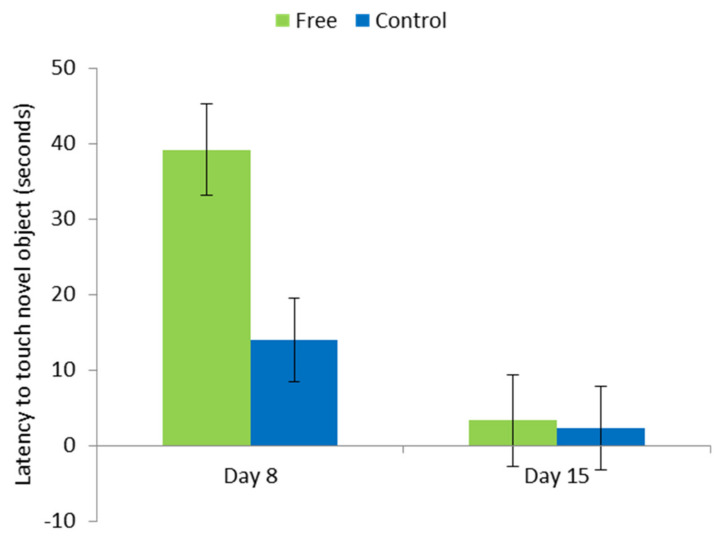
Latency for the first pig to touch the novel object at 8 and 15 days post weaning. There was an effect of treatment (*p* < 0.05), and an interaction between treatment and examination time (*p* < 0.05). ‘Free’ indicates pigs which had been reared in pens where the sow was free to turn around from day 4 of lactation to weaning and ‘control’ conventional farrowing crates (Crate).

**Table 1 animals-11-01181-t001:** Ethogram used to record piglet behaviours during lactation, adapted from Martin, Ison and Baxter [15].

Category	Behaviour	Description
Locomotory play	Scamper/Run	Two or more forward directed hops, running in a forward motion
	Pivot	Twirling around of the body, at least 90°
	Hop	Either two front feet or all four feet off the floor
Social	Nudge	Snout used to touch another piglets body
	Chase	Running after another piglet who is also running
	Social interaction/play	Sniffing, nuzzling by a piglet of another piglets head, face, nose
	Sow climb	Minimum of two feet off the floor and on the sow, not directed towards udder, climbing over udder or on sows head, neck, shoulders
	Sow nudge	Snout used to gently touch sows body
	Sow interaction	Sniffing, nuzzling sows head, nose
Object	Pen	Rooting, biting, sniffing or any other oral behaviour directed to pen fixtures or the crate
	Enrichment	Rooting, biting, sniffing, or any other oral behaviour directed to enrichment materials
Damaging	Ear biting	Ear biting
	Tail biting	Tail biting
Fighting		Forceful pushing or biting of another piglet

**Table 2 animals-11-01181-t002:** Piglet hoof scoring adapted from Lewis et al. [29].

Score	Description	
0	No damage	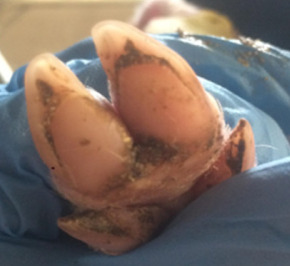
1	Mild bruising	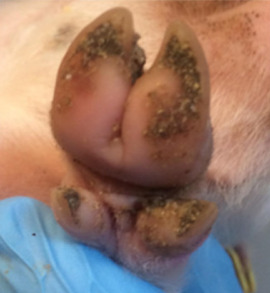
2	Severe bruising and/or small cut(s)	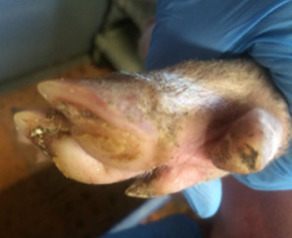
3	Large cut(s) and/or swelling	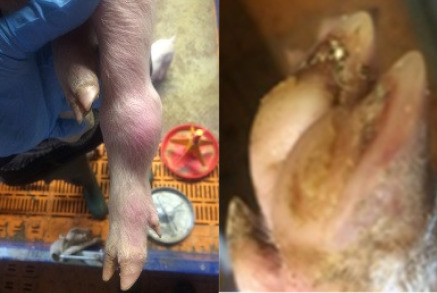
4	Hoof deformed/partially or fully amputated	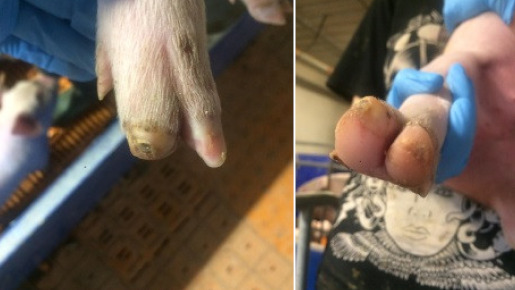

**Table 3 animals-11-01181-t003:** Mortality data for piglets reared in free lactation crates (free) and conventional farrowing crates (control) prior to weaning.

Measurements	Control	Free	*p*-Value
Initial litter size ^1^	14.79 ± 1.61	14.64 ± 1.47	NA
% mortality	14.42 ± 2.15	15.95 ± 2.31	0.61
% weaned	85.58 ± 2.15	84.05 ± 2.31	0.61
Day of death ^2^	8.69 ± 1.31	6.14 ± 1.65	0.43

^1^ Unanalysed mean ± standard deviation provided. ^2^ Least squares means were calculated by running raw data through the model, and *p*-values by running log transformed data.

**Table 4 animals-11-01181-t004:** Causes of death in each treatment before and after crate opening (morning of day 4 post farrowing) in the free and control treatments. Numbers of piglets that died due to each cause before and after were compared across treatments using Fisher’s exact test.

Cause of Death	Control	Free	*p*-Value
**Crushing**			
Before D4	21	17	0.017
After D4	9	26	
**Hunger**			
Before D4	0	1	0.214
After D4	11	2	
**Euthanasia**			
Before D4	8	6	0.649
After D4	6	2	

## Data Availability

The data presented in this study are available on request from the corresponding author.

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
