# Peer review of "Freedom to Grow: Improving Sow Welfare also Benefits Piglets"

_animals, 2021, doi:10.3390/ani11041181_

Round 1

Reviewer 1 Report

General comments

This study is looking at effects on piglet behaviour and performance in a farrowing pen with increased space allowance and temporary crating of the sow, as compared to a traditional pen. Several beneficial effects are identified. The study contributes to a field with high current interest and an urgent need for information to base practical recommendations on, that is, a decrease in the use of farrowing crates for sows. The paper holds a high standard and is well written. I have some comments on the documentation of a few details, as well as on the interpretation of the results.

Throughout the Simple Summary, Abstract, Discussion and Conclusion the reader gets a feeling that the observed effects of the treatment are attributed to freedom of the sow during lactation. This was, however, only part of the treatment which also included short-time freedom pre-farrowing for the sow, and a 22% larger pen area available for the piglets. In my opinion these features should not be overlooked as causes for or contributors to the observed effects. The most problematic part of the paper in this regard is L 568-570 in the discussion, where freedom during lactation is stated to be the cause for the observed effects. I suggest changing this sentence to include the possible influence of the other features of the treatment.

The emphasis on freedom during lactation is probably mostly unintentional. In the Abstract, for example, the treatment is introduced as “free lactation pen”. However, as no other information on the treatment is provided, the reader may be left with the belief that that freedom during lactation was the treatment of interest. It could be argued that the reader should associate free lactation with increased space allowance, however, similar experiments are carried out without differences in pen area. Thus I suggest to keep the reader aware of all the features of the treatment. I also suggest adding some thoughts on the possibility that space for the piglets and/or pre-farrowing freedom would cause or contribute to the observed effects.

Specific comments

For piglets, “hunger” was a reason for removal from the study. The term is mentioned a couple of times before it is defined at L 313. I suggest moving the definition to the first time it is mentioned

L 93-, 226-, 597- fecal cortisol

Is there any information on possible degradation of cortisol in fecal samples? May the age of the sample have an effect on the concentration (although a fresh sample was chosen)?

A sample of 5 g appears quite small. Were attempts made to include feces from more than one animal? The information in the text is not completely clear: L 228 gives an impression of a sample from one spot: “A fresh faecal sample…. was collected from the dung pile in the corner of the pen”), whereas L 600-601 (“the sample was taken on pen level”) could indicate several subsamples. Is it possible that it included feces from one individual only? According to the discussion the researchers assumed that this could be the case.

Table 1. L 223

“Locomotion” includes only certain forms of moving. Why did you not include simple walking or running, which most probably are very frequently occurring activities? I also suggest adding a note in the discussion on which forms of locomotion are included, just to remind the reader that it is somewhat unconventional.

Fighting: I suggest including who the behaviour was directed at (littermate?). How was “aggressive” defined? Would it not be more clear to say just pushing or biting of littermate (?).

L 255-, 573- Growth and feed conversion ratio

The amount of delivered feed was considered to equal the amount of ingested feed. Was there no possibility of spillage, or was it considered to be negligible or constant across pens? I suggest including a note on how (possible) spillage was handled in the trial and analyses.

Could the observed differences in feed conversion ratio not be caused by more spillage in “free” groups? A larger spillage could result from playing with the food, especially as it was in the form of pellets and provided ad libitum.

L 483, Fig 6A, caption: I believe “pigs” should be plural in “free pig were…”

L599-601 Opening of the crate has been associated with short-time increased restlessness or stress in the sow, as well as increased chasing of piglets (Goumon et al 2018, Illmann et al 2020) – this could increase the stress experienced by the piglets.

Author Response

Many thanks for your review of our manuscript.

Response to Reviewer 1 Comments

General comments

Point 1: Throughout the Simple Summary, Abstract, Discussion and Conclusion the reader gets a feeling that the observed effects of the treatment are attributed to freedom of the sow during lactation. This was, however, only part of the treatment which also included short-time freedom pre-farrowing for the sow, and a 22% larger pen area available for the piglets. In my opinion these features should not be overlooked as causes for or contributors to the observed effects. The most problematic part of the paper in this regard is L 568-570 in the discussion, where freedom during lactation is stated to be the cause for the observed effects. I suggest changing this sentence to include the possible influence of the other features of the treatment.

The emphasis on freedom during lactation is probably mostly unintentional. In the Abstract, for example, the treatment is introduced as “free lactation pen”. However, as no other information on the treatment is provided, the reader may be left with the belief that that freedom during lactation was the treatment of interest. It could be argued that the reader should associate free lactation with increased space allowance, however, similar experiments are carried out without differences in pen area. Thus I suggest to keep the reader aware of all the features of the treatment. I also suggest adding some thoughts on the possibility that space for the piglets and/or pre-farrowing freedom would cause or contribute to the observed effects.

Response 1: Increased space allowance in free pens and sow freedom pre-farrowing have been highlighted to keep the reader aware of all features of the Free treatment, L 11, 22, 571 – 575, 638, 642.

This section in the discussion now includes the possible influence of increased space allowance pre-weaning on the observed effects, L 571 – 575.

Specific comments

Point 2: For piglets, “hunger” was a reason for removal from the study. The term is mentioned a couple of times before it is defined at L 313. I suggest moving the definition to the first time it is mentioned

Response 2: Definition moved to first mention of hunger, L 166.

Point 3: L 93-, 226-, 597- fecal cortisol

Is there any information on possible degradation of cortisol in fecal samples? May the age of the sample have an effect on the concentration (although a fresh sample was chosen)?

A sample of 5 g appears quite small. Were attempts made to include feces from more than one animal? The information in the text is not completely clear: L 228 gives an impression of a sample from one spot: “A fresh faecal sample…. was collected from the dung pile in the corner of the pen”), whereas L 600-601 (“the sample was taken on pen level”) could indicate several subsamples. Is it possible that it included feces from one individual only? According to the discussion the researchers assumed that this could be the case.

Response 3:  We have not found any information on possible degradation of cortisol in fecal samples. However, the samples chosen were fresh and so this should not be an issue.

Every effort was made to collect samples containing faeces from several piglets (ie. collecting parts of the sample from different locations in the dung pile) however it is still possible that a sample may have come from one individual only. The sampling method has been described more clearly, L 231 - 233.

Point 4: Table 1. L 223

“Locomotion” includes only certain forms of moving. Why did you not include simple walking or running, which most probably are very frequently occurring activities? I also suggest adding a note in the discussion on which forms of locomotion are included, just to remind the reader that it is somewhat unconventional.

Fighting: I suggest including who the behaviour was directed at (littermate?). How was “aggressive” defined? Would it not be more clear to say just pushing or biting of littermate (?).

Response 4: This category has been renamed as locomotory play to better describe the behaviours measured, L 227, Table 1.

Fighting: ‘Aggressive’ was defined as forceful pushing/biting, the table has been amended to reflect this and that the behaviour was directed toward another piglet, L 227, Table 1.

Point 5: L 255-, 573- Growth and feed conversion ratio

The amount of delivered feed was considered to equal the amount of ingested feed. Was there no possibility of spillage, or was it considered to be negligible or constant across pens? I suggest including a note on how (possible) spillage was handled in the trial and analyses.

Could the observed differences in feed conversion ratio not be caused by more spillage in “free” groups? A larger spillage could result from playing with the food, especially as it was in the form of pellets and provided ad libitum.

Response 5: Spillage was considered to be constant across pens, and was minimal – there was a rubber mat in front of the feeder which captured any spilled feed, and there was rarely anything on it. The design of the single space feeder is such that the potential for waste is minimised as there is a lip which holds in the feed. The feeder styles were all the same for both Free and Control piglets through the weaner and finisher stage, so they had the same level of access to it, so we considered potential for wastage to be similar across treatments

Point 6: L 483, Fig 6A, caption: I believe “pigs” should be plural in “free pig were…”

Response 6: Corrected to pigs, L 485.

Point 7: L599-601 Opening of the crate has been associated with short-time increased restlessness or stress in the sow, as well as increased chasing of piglets (Goumon et al 2018, Illmann et al 2020) – this could increase the stress experienced by the piglets.

Response 7: Noted that the sow becomes more active when the crate is opened and this could cause increased stress in piglets, L 609 - 613.

Reviewer 2 Report

The paper is well written. The study design is interesting although to follow 46 litters from birth to market, with intense observation during the suckling period is a great deal of work. The objectives are well described. 

It is not clear how the sample size was determined. If I understand the methods, litter group was the experimental unit from birth to market and after weaning there were only 12 or 13 litters in each treatment group which doesn't seem to be enough to draw conclusions about growth and feed efficiency?

My other concern is the possibility of bias in the interpretation of the results in favour of the Free group. Even the name suggests this group should be superior. I think the data showed litter weights were the same for both treatment groups, mortality was similar, although crushing increased after day 4 in the Free group. Behaviour was similar although piglets in the Free treatment group had more space and therefore more opportunity for play. I didn't do the space calculations but it appeared that the rooms with conventional crates have 7 farrowing places and the rooms with larger Free crates farrow 6 sows. Some discussion of the economic loss this represents is warranted. I think it is also worth discussing that it is difficult to measure the benefits the piglets derive from the sow having more freedom versus the piglets having more room as well. In general, because many of the statistical analysis results were marginal as far as significance, I would recommend more caution in the interpretation of the data and I would suggest more discussion of alternative interpretations and the limitations of the study.

Author Response

Many thanks for your review of our manuscript.

Response to Reviewer 2 Comments

Point 1: It is not clear how the sample size was determined. If I understand the methods, litter group was the experimental unit from birth to market and after weaning there were only 12 or 13 litters in each treatment group which doesn't seem to be enough to draw conclusions about growth and feed efficiency?

Response 1: Sample size calculations were carried out prior to the experiment starting, and were based upon behaviour measures in the crate. From this we determined the sample size of 24 sows per treatment for the pre-farrowing measurements. We intended to follow all litters to slaughter, but we thought it important to remove the confounding effect of pig number within the pen, which could have affected growth; we could not control for this at assigning the piglets to pens at weaning as we decided not to mix piglets from different litters to create equal pen numbers, as the resultant aggression could also confound the work; see Camp Montoro et al., 2021, which investigated effects of mixing and group size on growth to finish; https://doi.org/10.1186/s40813-020-00187-7. We carry out a lot of nutrition based research in the pig unit, and normally have 10 -12 pens per treatment (as in Camp Montoro et al., 2021), and considering we found statistical differences in the growth measurements, which align with our hypotheses, and follow on from improved growth in the Free treatment prior to weaning, we are confident that we had sufficient power to detect treatment differences.

Point 2: My other concern is the possibility of bias in the interpretation of the results in favour of the Free group. Even the name suggests this group should be superior. I think the data showed litter weights were the same for both treatment groups, mortality was similar, although crushing increased after day 4 in the Free group. Behaviour was similar although piglets in the Free treatment group had more space and therefore more opportunity for play. I didn't do the space calculations but it appeared that the rooms with conventional crates have 7 farrowing places and the rooms with larger Free crates farrow 6 sows. Some discussion of the economic loss this represents is warranted. I think it is also worth discussing that it is difficult to measure the benefits the piglets derive from the sow having more freedom versus the piglets having more room as well. In general, because many of the statistical analysis results were marginal as far as significance, I would recommend more caution in the interpretation of the data and I would suggest more discussion of alternative interpretations and the limitations of the study.

Response 2:  There was no intention to imply the treatment group as superior. The word 'free' is used as it is readily understood in terms of the experimental treatment. Treatments were similar in most respects, demonstrating an alternative management strategy to fully confining sows during farrowing and lactation which does not affect pre-weaning piglet mortality.

Piglets in free pens did have more space (5.5 vs. 4.6 m2) this has been highlighted and identified as a possible benefit, L 573 – 575.

The need to assess economic effects due to reduced number of farrowing places has been mentioned, L 575 - 580. We intend to submit the data to the Moorepark Pig Production Model, which will capture the effects you mentioned, so that we can try to fully evaluate the economic impacts of adopting Free lactation pens.